# The Reign of Follistatin in Tumors and Their Microenvironment: Implications for Drug Resistance

**DOI:** 10.3390/biology13020130

**Published:** 2024-02-19

**Authors:** Jennifer Sosa, Akinsola Oyelakin, Satrajit Sinha

**Affiliations:** 1Department of Biochemistry, Jacobs School of Medicine and Biomedical Sciences, University at Buffalo, Buffalo, NY 14203, USA; jsosa3@buffalo.edu; 2Center for Integrative Brain Research, Seattle Children’s Research Institute, Seattle Children’s Hospital, Seattle, WA 98101, USA; akinsola.oyelakin@seattlechildrens.org; 3Ben Towne Center for Childhood Cancer Research, Seattle Children’s Research Institute, Seattle Children’s Hospital, Seattle, WA 98101, USA

**Keywords:** follistatin, transforming growth factor-beta pathway, squamous cell carcinoma, tumor heterogeneity, chemoresistance, tumor microenvironment, cancer hallmarks

## Abstract

**Simple Summary:**

Within the extracellular milieu surrounding cancer cells exists an ecosystem of heterogeneous cell populations that dictate tumor growth and survival via cell–cell signaling induced by secreted factors. The transforming growth factor-β (TGF-β) superfamily, a crucial programmer of the tumor microenvironment, presents challenges as a therapeutic target due to its biphasic effects in cancer. Despite concerted efforts, the clinical efficacy of its antagonists remains elusive, reflecting a limited understanding of how cancer cells can become nonresponsive to the potent cytostatic effects of these ligands while retaining their tumor promoting effects. Follistatin, a secreted glycoprotein and an endogenous bioneutralizer of TGF-β ligands, has gained prominence and emerged as a potentially targetable modulator of key resident cell populations in the tumor microenvironment, particularly in squamous cell carcinoma. We delve deeper into the expression patterns and mechanistic role of follistatin in cancer, examining its intimate relationship with TGF-β and its implications in drug resistance across lung, ovarian, and head and neck cancers.

**Abstract:**

Follistatin (FST) is a potent neutralizer of the transforming growth factor-β superfamily and is associated with normal cellular programs and various hallmarks of cancer, such as proliferation, migration, angiogenesis, and immune evasion. The aberrant expression of FST by solid tumors is a well-documented observation, yet how FST influences tumor progression and therapy response remains unclear. The recent surge in omics data has revealed new insights into the molecular foundation underpinning tumor heterogeneity and its microenvironment, offering novel precision medicine-based opportunities to combat cancer. In this review, we discuss these recent FST-centric studies, thereby offering an updated perspective on the protean role of FST isoforms in shaping the complex cellular ecosystem of tumors and in mediating drug resistance.

## 1. Introduction

Over the years, substantial efforts have been dedicated to comprehending the role of genetic events, such as mutations, gene amplifications, and viral genome integration, in propelling malignant cellular states. However, with the advent of single-cell RNA-seq (scRNA-seq), proteomics, and spatial transcriptomics (ST), there is increased appreciation for the critical influence of cell–cell signaling, tumor architecture, and the secretome in driving disease etiology, progression, and resistance to therapy. Indeed, it is becoming clear that dynamic signaling cross-talk between cancer cells and the tumor microenvironment (TME) can drive and sustain tumor heterogeneity, enabling clonal evolution and selection for drug-resistant cell populations. These findings explain the low response rates exhibited by many cancer patients to immuno- and chemotherapies, and highlight the need for biomarkers that predict patient response as well as adjuvants that increase the efficacy of current therapies. In this context, the growth factor follistatin (FST) has garnered considerable attention and interest as an intriguing target across various cancers, prompting the need to explore its oncogenic role in the tumor milieu and the FST-dependent molecular events that govern cancer biology. In this review, we describe the structure, function, and expression profile of FST; its emerging role in governing key hallmarks of cancer; and highlight recent findings related to FST in the context of drug resistance and cancer progression in a number of solid tumors.

## 2. Follistatin and Its Intimate Tango with TGF-β

FST is a secreted single-chain glycoprotein that irreversibly binds and inhibits a spectrum of transforming growth factor-β (TGF-β) superfamily ligands, including activins, myostatin, TGF-βs, and bone morphogenic proteins (BMPs) [1]. Activins and BMPs signal via type I and type II serine/threonine kinase receptors that, upon dimerization and phosphorylation, promote the activation and nuclear translocation of receptor-associated Smads [2]. Downstream activin and TGF-β signaling is mediated via Smad2/3 proteins, while BMPs activate the Smad1/5/8 pathway, with Smad4 being the common mediator shared between both pathways for facilitating Smad nuclear translocation [3]. Members of the TGF-β family can also activate Smad-independent pathways such as the extracellular signal-regulated kinase/p38 mitogen-activated protein kinase (ERK/MAPK) and phosphatidylinositol-3-kinase/AKT pathways to promote cell growth and survival [4,5,6,7]. FST neutralizes these signals by binding TGF-β members in a 2:1 stoichiometry, occluding types I and II receptor binding domains [8,9]. While the inhibitory effect of FST on the canonical Smad pathway is well documented [10], its effects on the noncanonical pathways is less clear, with some reports indicating that FST may potentiate these pathways independently of TGF-β signaling [10,11,12]. Given that there is a broad range of TGF-β signaling responses resulting from the integration of different signals from myriad ligands and receptor combinations [13,14,15], it is likely that the varying affinities of FST for different TGF-β members could have distinct biological outcomes. This is particularly relevant in scenarios where the pleiotropic effects of TGF-β signaling are prominent, such as in cancer progression and therapy [16].

## 3. Complexity of Follistatin Isoforms and Structure Can Be Exploited for Targeted Therapy

The *FST* gene comprises six exons that give rise to two precursor mRNA molecules via alternative splicing: pre-FST344 and pre-FST317 [17]. These yield the two major isoforms, FST315 and FST288, respectively, following post-translational cleavage of the signal peptide (Figure 1A). The longer isoform contains five domains: the N-terminal domain, three cysteine-rich FST domains (FSD1–FSD3), and the intrinsically disordered C-terminal domain [18]. The C-terminal domain comprises an acidic carboxyl tail that interacts with a basic site within FSD1. This interaction prevents the binding of heparin and heparin sulfate (HS) to apo-FST315 and thus blocks the localization of FST to cell surfaces [19,20]. Consequently, FST315 is the major circulating isoform [21]. However, this weak affinity for cell surface HS is strengthened in the presence of TGF-β, enabling HS to play a role in TGF-β clearance from circulation [20,22,23]. Proteolytic processing of the C-terminal domain of FST315 results in a third isoform, FST303, that is predominantly found in follicular fluid [24]. The shorter FST isoform, FST288, lacks an acidic carboxyl terminus, allowing it to freely associate with HS on the cell membrane, where its effects are localized [1,19,21].

The cysteine-rich domains of FST contain epidermal growth factor-like and kazal serine protease inhibitor-like subdomains connected by a hinge region, enabling FSD1 and FSD2 to “wrap” around TGF-β ligands, thereby blocking cognate receptor binding sites [9,23,25] (Figure 1B). Although not essential for activin neutralization, FSD3 confers ligand affinity and contributes to the 2:1 FST–ligand complex formation [26,27,28]. The structural knowledge and the innate ability of FST to sequester TGF-β ligands extracellularly has been exploited to design potent and localized derivatives. Such FST-based biologics, often designed as a fusion protein, have shown therapeutic promise in the treatment of various conditions, such as myopathies that afflict skeletal muscle. [29,30]. Despite these interesting forays into the biochemical and molecular aspects of FST, significant gaps persist in our understanding of the protean nature of FST, warranting further investigation. Specific aspects of FST that deserve follow-up studies include its interactions with BMPs, which exhibit lower binding affinities in contrast to activin and myostatin [1], as well as how the various isoforms engage with other proposed partners, such as angiogenin and fibronectin [31,32].

## 4. Post-Translational Modification and Intracellular Localization of Follistatin

The intricate nature of FST isoforms and their structural complexity is paralleled by their post-translational modifications and localization to distinct intracellular compartments. The N-terminal domain of FST is preceded by a signal peptide containing two methionine residues (Figure 2). Early studies found that the mutation of the first methionine abolishes any detectable protein synthesis, whereas complete deletion of the signal peptide yields a functional FST protein [31,33,34]. Notably, follistatin-like 3 (FSTL3), which is structurally related to FST, also has two methionines: one in the signal peptide region and one that acts as an alternative translational start site within the start of the N-terminal domain [34]. The translation of FSTL3 from the second methionine results in a bioactive protein with nuclear retention that lacks glycosylation and remains intracellular [34]. This raises the tantalizing possibility that the omission of the signal peptide, under physiological or pathological conditions, can lead to varied glycosylation patterns in FST, consequently influencing its cellular localization and function. Given the limited studies on FST biosynthesis, it will be important to investigate the possibility of alternative translation start sites, especially since both glycosylated and non-glycosylated forms of FST have been identified [35].

FST can be glycosylated at three putative sites: Asparagine 95, 112, and 295 [19,35,36]. The various glycosylated forms of FST appear to exhibit comparable activin binding affinity, yet their glycosylation status may hold the key to additional, yet unknown biological roles [19,34]. Notably, the glycosylation state can significantly influence the stability of FST, as evident from the enhanced pharmacokinetic properties observed in FST-based therapies with increased glycosylation [36]. Furthermore, glycosylation status also appears to impact the secretion of FST isoforms, with secreted variants displaying heterogeneous glycosylation patterns in contrast to their intracellular counterparts [34].

The discovery of FST within the nucleus, similar to what has been reported for FSTL3, adds a novel dimension to its functional role [33]. In the nucleus, the enrichment of FST was found to be in the nucleolar compartment, a hub for ribosomal biogenesis, and interestingly associated with glucose deprivation of cervical cancer-derived HeLa cells [33]. The presence of a nuclear localization signal (NLS) in the FST protein has been shown to mediate this process, resulting in reduced rRNA synthesis. In agreement with these observations, nuclear staining for FST has also been observed in cervical intraepithelial neoplasia (CIN), cervical squamous cell carcinoma (CSSC), thymomas, and thymic carcinomas [37,38]. The results from the CIN and CSCC studies are particularly revealing since they show overall (a) higher nuclear immunoscores for FST in the control tissues with no lesions and (b) decreased nuclear immunoscores in a cellular-differentiation-dependent manner with progression from CIN lesions to cancer. While the aforementioned tumor staining studies per se do not provide high-enough resolution to pinpoint the possible nucleolar localization of FST, it is tempting to speculate the possibility that decreased nuclear shuttling of FST relieves its repressive effects on rRNA transcription, allowing cancer cells to exploit rRNA biogenesis [33]. Another noteworthy consideration is that studies on the nuclear localization of FST were performed using FST expression constructs lacking the signal peptide, making it plausible that the nuclear retention of FST is dependent on the post-translational glycosylation status of FST. Currently, there is no information about the nature of FST mRNA isoforms, glycosylation, or other modification statuses of the FST protein that resides in the nucleolus/nucleus. These unanswered questions set the stage for future exploration regarding the complex patterns of the subcellular localization of FST, particularly in the broader context of cell stress and oncogenesis.

## 5. Expression of Follistatin in Tissues and Organs

Although FST is expressed ubiquitously at varying levels in almost all tissues, early studies were mostly focused on a small number of organs, such as the gonads, pituitary gland, and muscles, to name a few. The generation of bulk and scRNA-seq-based databases such as the Genotype-Tissue Expression (GTEx) Portal and The Human Protein Atlas (TCGA) has shed new light onto the expression profile of FST. Indeed, two distinct and interesting features of *FST* expression are evident from the analysis of such tissue transcriptomic data. First, *FST* is certainly widely expressed; however, there is a broad range of expression levels, at least based on transcript-per-million values. The top five tissues with the highest levels of *FST* expression in order are the liver, bladder, adipose tissue, placenta, and skin. While some of these organs have served as good model systems for studying FST biology (e.g., the liver and adipose tissue), others, such as the bladder, placenta, and skin, remain somewhat unexplored. In agreement with the high levels of tissue mRNA expression, circulating FST has been shown to be primarily derived from the liver and adipose tissues, where its secretion is intimately tied to imbalances in energy metabolism [11,39,40,41,42]. The second, perhaps more telling, revelation regarding *FST* expression comes from the scRNA-seq analysis of human tissues. Here, paralleling the bulk tissue patterns, *FST* expression is found to be high in skeletal myocytes, fibroblast mesenchymal cells, granulosa cells, and hepatocytes.

Surprisingly, the next-ranked group of *FST*-expressing human cells are epithelial in nature, representing salivary ductal cells, basal respiratory cells, myoepithelial cells, and keratinocytes. An examination of Tabula Muris, a compendium of single-cell transcriptome data from 100,000 cells from 20 mouse organs and tissues, shows similar trends, with basal cells of the mammary gland and keratinocyte stem cells of the skin being the top two *FST*-expressing cell populations. In agreement with this, recent studies on the potential role of FST in stem and progenitor cells of mouse tissues such as the skin [43,44,45] and the salivary gland [46] point at an overlooked epithelial-centric role of FST. The similar patterns of expression of *FST* between mice and humans suggest a functional conservation which aligns well with the fact that the FST protein is remarkably identical (~98% at the amino acid) between the two species (Figure 2). A similar high conservation of TGF-β amino acid sequences and signaling pathways between humans and mice [47] offers high confidence that mouse models employed in studying FST biology are biologically relevant and hold clinical translational significance. Indeed, studies of muscular dystrophy in mouse models using human recombinant FST or the human *FST* transgene show cross-species activity between human FST and mouse tissues [48,49,50,51].

The relative expression levels of the FST isoforms under normal and pathological conditions is another aspect of FST biology that needs closer scrutiny; however, prior studies, albeit limited in scope, suggest that they are generated at different rates, with FST288 often less than a tenth of FST315 [21,52,53]. Since evidence suggesting different biological functions for FST isoforms is limited [1], it is worth probing whether the relative abundance of FST isoforms change under conditions of stress or disease and how this shift ultimately influences molecular and cellular outcomes. 

## 6. The TGF-β Paradox and the Possible Role of Follistatin

TGF-β has dual roles in cancer, often referred to as the ‘TGF-β paradox’, in which it acts as a tumor suppressor in early-stage cancer and as a tumor promoter in later stages [54]. This paradox has been attributed to mutations or the loss of TGF-β receptors, rendering cancer cells unresponsive to TGF-β but responsive to the tumor-promoting effects of TGF-β on immune and stromal components [55]. Mutations in the TGF-β pathway are rare, but the downregulation of Smad signaling, often associated with reduced TGF-β type II receptor expression, is linked to the malignant transformation of squamous cells [56,57,58]. It is also possible that signals within the TME similarly reduce TGF-β activity in tumor cells or establish a state of TGF-β resistance. Notwithstanding the inherent complexity, TGF-β remains a key therapeutic target. Indeed, the TGF-β family member activin has been targeted to limit tumor growth, dissemination, and drug resistance [59,60], though activin inhibitors have yet to demonstrate anti-tumor efficacy [61].

Elevated serum FST is a common feature in various cancers. For instance, higher serum FST levels correlate with bone metastasis in prostate cancer and are associated with larger tumor sizes and a poorer prognosis in patients with hepatocellular carcinoma [62,63]. Serum FST is also associated with tumor stage in lung and advanced colorectal cancers [64,65]. Remarkably, it has been proposed that the root cause of dysregulated serum FST levels might stem from the tumors themselves, as surgical resection of thymic epithelial tumors results in the baseline resetting of FST levels within patient serum [38]. Indeed, FST is consistently found to be secreted by solid tumors, including melanomas, ovarian adenocarcinomas, and lung adenocarcinomas, and its expression exhibits a notable upregulation when compared to normal tissue samples [66,67,68].

These observations raise the possibility that FST contributes to the dichotomous nature of TGF-β signaling in cancer by promoting resistance to the potent cytostatic effects of TGF-β in tumor cells. It is also conceivable that, akin to TGF-β, FST also exhibits biphasic effects. This potential dual role may serve to counterbalance the pronounced effects of TGF-β within the TME [69]. High-throughput technologies, such as proteomics, scRNA-seq, and spatial transcriptomics (transcriptomic profiling of individual cells within their native tissue architecture), have begun to unravel our understanding of the role of FST in cancer as a potent inhibitor of the TGF-β pathway. Further in-depth examination of the dynamics between FST and TGF-β in oncogenesis remains a critical research area.

Given the complex biochemical properties of FST, its function in crucial developmental processes, and its intimate association with TGF-β signaling, it is not surprising that a major role for FST in various aspects of tumor biology and treatment has been unearthed recently. We delve into this rapidly unfolding literature in the next segments by exploring FST’s involvement in fostering crucial oncogenic processes, such as stem cell renewal, proliferation, metastasis, and angiogenesis, and highlight FST’s emerging role in mediating drug resistance in lung, ovarian, and head and neck cancers. Finally, we offer a snapshot of evolving molecular mechanisms and signaling pathways that control the aberrant expression of FST in cancer cells.

## 7. Follistatin in the Ecology of Cancer: Cancer Hallmarks

### 7.1. Maintenance of a Progenitor State in Epithelial-Rich Tissues

Since TGF-β signaling induces the differentiation of basal epithelial cells, FST may serve to maintain cells in a progenitor state by inhibiting TGF-β [70,71]. Accordingly, as basal cells differentiate, such as in the case of skin keratinocytes, *FST* expression is downregulated [72,73]. Several recent studies provide additional evidence for a role of FST in stem cell maintenance across different systems, including taste bud progenitors, planarian neoblast cells, Drosophila hub cells, murine auditory pro-sensory cells, and thymic epithelial progenitor cells [74,75,76,77,78]. Additionally, we have shown that the transcription factor p63, known to maintain the pool of keratinocyte progenitors in the basal and myoepithelial epithelium, orchestrates a stem/progenitor state through the FST-dependent inhibition of TGF-β signaling in salivary gland epithelial cells [46]. Given the convergence of p63 and FST expression in the basal resident stem and progenitor cell compartment of the epithelium, it is likely that the roles of these two molecules may be intertwined, not only in normal biology, but in tumor biology as well. Finally, it is worth mentioning that activation of the TGF-β pathway occurs under stress conditions in differentiating squamous cells [71]. This holds significance, as *FST* expression not only rises in response to diverse stress conditions, including hypoxia, radiation, glucose deprivation, and oxidative stress, but has also demonstrated a protective role against oxidative damage [33,79,80,81,82]. These findings suggest that FST potentially inhibits TGF-β signaling to sustain basal cells in a quiescent state and counteracts stress-induced differentiation of the basal epithelium.

### 7.2. Sustaining the Growth Machinery

Gain- and loss-of-function studies in various epithelial cancers indicate that FST enhances cell proliferation and acts primarily in a pro-tumorigenic fashion [83,84,85,86]. However, given the intimate link between FST and TGF-β and the dual role of TGF-β acting as both tumor suppressor and promoter, it is not surprising to find conflicting outcomes upon FST dysregulation.

For example, in mammary carcinoma cells, activin promotes proliferation, while FST reduces tumor growth by enhancing cellular apoptosis [87]. In contrast, in prostate cancer, activin inhibits proliferation and induces apoptosis [88]. FST may thus be protective in breast cancer but oncogenic in prostate cancer. Indeed, FST has been described as a good predictor of survival in breast cancer, with high serum FST correlating with better overall and relapse-free survival [89] and low FST levels associated with increased metastasis and reduced survival [90]. Furthermore, breast cancer tissues with low *FST* expression exhibit increased proliferation, migration, and invasion [91]. However, the overexpression of *FST* in human breast adenocarcinoma cells increases proliferation but reduces invasion, predicting better survival [92]. This hints at the possibility that serum and tissue *FST* expression levels may differ in their oncogenic and predictive outcomes and that the effect of FST on cell proliferation may be context-dependent.

### 7.3. Cancer Metastasis

Activated fibroblasts and myofibroblasts are the primary synthesizers and mechanical regulators of the extracellular matrix (ECM). The activation of myofibroblasts is predominantly mediated by TGF-β, which, in cancer, is actively secreted by tumor cells [93]. Tumor-associated myofibroblasts, or cancer-associated fibroblasts (CAFs), support tumor growth and dissemination by remodeling the ECM [94,95]. Single-cell analysis has unveiled the diversity of CAF populations, with spatial distribution being a key determinant of CAF functional states [96,97]. However, our understanding of how FST affects the function of CAFs remains limited. We do know that CAFs secrete FST, as documented in human tumor tissues and cultured systems [98,99]. Notably, exogenously applied FST enhances the invasion of dysplastic epithelial cells in organotypic cultures, even in the absence of fibroblasts [100]. In contrast, co-culture models have shown that FST secreted from both cellular compartments supports CAF-mediated metastasis and cancer cell proliferation [98]. This also suggests an additive autocrine/paracrine effect by which FST derived from fibroblast and epithelial compartments binds and sequesters TGF-β molecules within the ECM, tipping the balance towards an invasive epithelial phenotype [100]. 

### 7.4. Angiogenesis

Angiogenesis, the formation of new blood vessels, is a crucial process for tumor growth, as it not only sustains tumor development but also offers an escape route for cancer cells to enter the bloodstream during dissemination. In patients with thymic epithelial tumors, there is a noteworthy positive correlation between serum FST levels and the density of mature tumor microvessels [38]. Furthermore, mature vascularization is closely linked to advanced tumor stage and invasiveness in thymic epithelial tumors [101]. This effect has also been observed in experimental models where xenografts derived from human R30C mammary carcinoma cells that overexpress FST exhibit tumors with higher microvessel densities compared to tumors overexpressing activin. However, the maturities of microvessels in both tumor types are comparable, though vessels in activin-overexpressing tumors are smaller in diameter [87]. 

The enhancement of angiogenesis by FST is amplified in the presence of angiogenic factors, such as hepatocyte growth factor (HGF) and fibroblast growth factor (FGF), which are primarily derived from stromal cells such as CAFs [102,103,104]. This finding is notable because tumor vascularization often involves the migration of endothelial cells triggered by the release of angiogenic factors from stromal cells [94,105]. Indeed, FST promotes microvascular network formation and capillary sprouting in three-dimensional spheroid assays by directly affecting endothelial cell migration [87,106]. Altogether, the findings described above indicate that FST is an important driver of the tumor ecosystem, directing the cross-talk between CAFs, endothelial cells, and epithelial cells to foster tumor growth and facilitate metastasis.

## 8. Follistatin in Drug Resistance: Vignettes from Three Major Cancer Types

### 8.1. Lung Cancer

Unlike normal fibroblasts, which play crucial roles in tissue repair and homeostasis, CAFs have been reprogrammed and activated by cancer cells to support tumor growth, progression, and drug resistance [107,108]. These outcomes are driven by heterogenous CAF populations with tumor-protective or tumor-suppressive functions. The spatial interplay between epithelial tumor cells and CAFs are vital in nurturing their symbiotic relationship. This interaction fosters a bidirectional signaling cascade, potentially leading to therapy resistance and enabling immune evasion in cancer cells [109]. Microarray analysis revealed a 21-fold increase in FST expression in lung carcinoma cells following direct contact with fibroblasts [110], highlighting a role of fibroblasts or fibroblast-secreted factors in the regulation of FST in cancer epithelial cells. While the role of FST in CAFs has largely been overlooked, its increased secretion, also observed in prostate epithelial cancer cells and CAFs upon direct contact [98], merits closer investigation.

Along with immunotherapy and platinum-based drugs, epidermal growth factor receptor–tyrosine kinase inhibitors (EGFR-TKIs) represent a cornerstone in standard-of-care interventions; however, resistance develops within months following treatment [111]. In non-small-cell lung cancer (NSCLC), specific CAF subsets have been linked to EGFR-TKI resistance [99]. These subsets secrete soluble growth factors, including FGF7 and HGF, which activate MAPK signaling in cancer cells, circumventing EGFR activation. Among the top five factors secreted by CAFs in lung cancer, FST surprisingly emerged as a critical player in rescuing cancer cells from EGFR-TKI-induced toxicity, and thus represents a potential target to combat drug resistance in NSCLC (Figure 3A). 

In a phase II clinical trial examining the effectiveness of an EGFR-TKI (Erlotinib) in combination with an antiangiogenic drug (bevacizumab), biomarker analysis revealed elevated levels of FST in the sera of patients subjected to combination therapy [112]. This was further associated with worse treatment outcomes and increased serum concentrations of angiogenic factors [112]. However, it is important to note that NSCLC is a heterogeneous disease including the more prevalent subtypes lung adenocarcinoma (LUAD) and lung squamous cell carcinoma (LUSCC) [113]. Therefore, the differential tumor compositional landscape may potentially influence the extent to which FST contributes to these cancer types. Notably, the inhibition of TGF-β signaling via systemic administration of FST increases the efficacy of carboplatin therapy in LUAD [114]. Indeed, LUSCC tumors exhibit a significant upregulation in FST expression compared to normal tissue, unlike in LUAD (Figure 4), suggesting FST may play a more significant role in this disease. However, the impact of this dysregulation on LUSCC progression remains unexplored. Therefore, determining whether targeting FST can improve the effectiveness of drug therapy in LUSCC requires careful validation using cancer subtype-specific models.

With the knowledge that growth factors induce FST expression in keratinocytes, it is essential to understand how the secretion of FST and such factors by CAFs contributes to the diminished responsiveness of cancer cells to TGF-β [115]. Considering that FST is not only elevated in the sera but also within the tumors of lung cancer patients, it is conceivable that the release of FST by CAFs forms a protective barrier around tumor cells, making them less responsive to TGF-β signals and ultimately promoting cell survival despite therapeutic pressures. Hence, targeting TGF-β to overcome drug resistance may not be a one-size-fits-all solution and would necessitate a comprehensive understanding of how various cell types within the TME respond to TGF-β. This may also involve developing strategies to restore TGF-β responsiveness in cancer cells. 

### 8.2. Ovarian Cancer

Cancer cells utilize several strategies to evade elimination. One such strategy is the expression of immune checkpoint molecules, such as PDL-1 and CTLA-4, that suppress anti-tumor T-cell responses and are the basis of the class of therapies called immune checkpoint inhibitors. In addition, cancer cells also secrete factors that influence the localization of pro- and anti-tumor immune infiltrates within the TME. A recent transcriptomic analysis of patient-derived xenografts showed that FST is upregulated in ovarian tumors resistant to combination chemotherapy with immune checkpoint inhibitors, whereas the genetic ablation of FST restored drug susceptibility and improved the overall survival of mice with these tumors [84]. One plausible mechanism by which FST may exert its resistance-promoting effects is by creating an immune-cold TME (Figure 3B), as suggested by the negative enrichment of genes associated with inflammatory responses in ovarian cancer cells overexpressing FST [84]. Inflammatory signals are crucial in mediating anti-tumor immunity by recruiting cytotoxic effector cells to the tumor site. The intriguing aspect here is that FST appears to suppress these inflammatory signals, which is unexpected considering that TGF-β, a known immune tolerance mediator associated with the restriction of T-cell infiltration, operates in a similar context.

While the precise mechanism by which FST dampens immune responses in ovarian cancer to facilitate immunotherapeutic resistance remains unclear, one plausible explanation is that its inhibition of TGF-β disrupts the adaptive immune responses necessary for identifying and targeting malignant cells. Indeed, FST has the ability to modulate immune responses in various contexts. For instance, FST diminishes cytokine production by T-helper 2 cells in the lung epithelium under conditions of chronic allergen challenge [116]. Additionally, FST has the capacity to decrease the abundance of neutrophils and macrophages in the airways of cystic fibrosis models, and it can also inhibit the differentiation of pathogenic-Th17 cells that play a crucial role in the pro-inflammatory induction of immune responses [117,118]. Conversely, FST has also been found to bolster anti-tumor immunity and block metastasis by inhibiting activin-A signaling in tumor-resident natural killer cells in an orthotopic melanoma mouse model [119]. These findings collectively suggest that FST may exhibit opposing roles in adaptive and innate immune responses, potentially carrying implications for the relationship between immunotherapy efficacy and the tumor immune microenvironment.

Recent work by Cole et al. has echoed these findings, suggesting that FST also promotes resistance to platinum therapy, albeit through a different mechanism. This group showed that FST is upregulated in response to cisplatin treatment in epithelial ovarian cancer cells. Furthermore, this response was accompanied by the induction of quiescence in neighboring cells and reduced apoptosis via the activation of the activating transcription factor 2, a member of the AP-1 transcriptional complex [120] (Figure 3C). Notably, these quiescent cell populations expressed putative cancer stem cell markers, ALDH and CD133, that are upregulated in recurrent tumors and associated with decreased drug responses [121]. Similar outcomes were also observed in xenograft models established from platinum-resistant cells depleted of FST. These models exhibited a substantial delay in the development of metastatic disease following paclitaxel treatment compared to cells with intact FST [120].

An analysis of intraperitoneal aspirates collected from chemotherapy-naïve ovarian cancer patients, both before and after undergoing cycles of treatment, indicated an immediate upregulation of FST levels following cisplatin treatment. Remarkably, FST levels dropped significantly to pre-treatment baseline levels three months after the completion of treatment [120]. This suggests that FST secretion may be part of an acute phase response to chemotherapy. However, the implications of dysregulated FST secretion following therapy remains unclear given the confounding factors presented in this phase I clinical trial, from which patient samples were obtained [122]. Additionally, given the limited sample size of these studies, there is a need for more detailed evaluation of the impact of chemotherapy or immunotherapy on FST expression and vice versa, using patient data obtained from such large-scale transcriptomics studies as the TCGA. Unfortunately, a substantial portion of the patient data within this resource originate from patients with treatment-naïve tumors or individuals with incomplete treatment annotation [123]. Hence, there is a notable absence of robust patient data that could shed light on the dynamic relationship between FST expression, therapeutic regimen, and patient response to therapies. Nonetheless, these initial findings hold significant implications as they suggest that the functions of FST extend beyond the realm of autocrine signaling and that FST may serve a protective function in the TME under drug-induced cellular stress.

### 8.3. Head and Neck Squamous Cell Carcinoma

Head and neck squamous cell carcinoma (HNSCC) represents a diverse cluster of cancers originating from epithelial cells lining the oral, nasal, and throat mucosa. Genomic instability, driven by HPV genome integration, and copy number alterations, resulting in the amplification of oncogenes such as *EGFR* and *TP63*, further complicate its diagnosis [124,125]. Given its molecular complexity and frequently delayed diagnosis, extensive efforts have been devoted to molecular classifications and unraveling the intricate signaling dynamics within the TME. We and others have proposed FST as a potential biomarker in HNSCC, underscoring its significance in the malignant transformation of the oral epithelium [83,126].

Recent spatial transcriptomic analyses have revealed two distinct domains within HNSCC tumors—the tumor core and the leading edge—each characterized by unique gene signatures [127]. The tumor core primarily relies on mesenchymal-to-epithelial transition paracrine signaling, while the leading edge recruits cells from the tumor core through epithelial-to-mesenchymal transition (EMT) signals and sustains itself through cross-talk with CAFs [127,128]. Notably, FST expression is enriched within the cells of the leading edge (Figure 5), an invasive niche usually characterized by immunosuppressive, migratory, and basal-stem-like features [129,130,131].

Spatial transcriptomics and scRNA-seq analyses of cutaneous squamous cell carcinoma similarly localized FST to the leading edge, specifically in a cell population termed “tumor-specific keratinocytes” [131]. Like the HNSCC tumor leading edge, these cells exhibit partial EMT features and serve as a “hub” for communication between the tumor-stroma. Cells undergoing EMT acquire characteristics reminiscent of cancer stem cells, including the ability to evade programmed cell death and enhance drug resistance through the upregulation of drug efflux pumps [132]. The activation of TGF-β-dependent signaling pathways primarily drives this phenotypic shift. However, clinical attempts to target TGF-β to inhibit EMT processes have faced significant challenges [132,133]. One notable obstacle in tackling EMT lies in its complex nature. Recent insights from scRNA-seq studies indicate that EMT is a multi-state phenomenon, and that cancer cells often exist within this transitional, hybrid, or intermediate state, contributing to the dynamic nature of tumors [128,134]. Given that FST is enriched within the invasive tumor front within cells that exhibit this partial EMT phenotype, FST likely plays a role in communicating metastatic signals, evading immune detection, and driving a cell survival program that limits therapeutic efficacy.

Transcriptomic studies have shown that cisplatin treatment upregulates FST within the cancer stem cell population of HNSCC [135], suggesting that comparable resistance mechanisms may be at play, as in ovarian cancer. However, as surgical resection of HNSCC tumors commonly precedes adjuvant therapy, there is a lack of understanding of the mechanisms that drive resistance. Nonetheless, to understand the early molecular events preceding late-stage tumor aggression, the dysregulation of specific genes should be utilized as valuable indicators of prognosis. In this context, microarray and RNA-seq studies consistently corroborate the upregulation of FST in HNSCC patients (Figure 4), which correlates with reduced survival outcomes [86,136,137]. Moreover, experimental models have provided additional evidence for FST in promoting cancer cell growth, colony formation, migration, and invasion [83,130]. 

## 9. Mechanisms Driving Follistatin Expression in Cancer

It is largely unknown what regulators alter FST expression levels under the different stress conditions reported in the literature [138]. Under basal conditions, it is widely acknowledged that FST is typically governed by a TGF-β-Smad-induced autoregulatory negative feedback loop (Figure 6A). However, questions persist about whether this mechanism is operational in epithelial cells and why there is a frequent co-occurrence of aberrant TGF-β and FST expression in cancer [10,139,140,141].

Considering that *FST* expression is regulated by TGF-β signaling and the likelihood that cancer cells often become unresponsive to TGF-β as oncogenesis progresses, it becomes apparent that additional mechanisms must come into play to sustain high levels of FST expression in cancer. To explore this, our group recently harnessed scRNA-seq data from HNSCC primary tumors to examine the origin of *FST* expression within the TME [126,128]. We found a notable enrichment of *FST* within *TP63*-expressing tumor epithelial cells driven by the activation of the MAPK signaling pathway (Figure 6B). As increased copy number aberrations often correlate with increased activity of the amplified gene [125,142], we posited that the amplification of *EGFR* and *TP63* in HNSCC may explain the upregulation of FST within tumor epithelial cells. While the complete mechanism remains unresolved, it is possible that super-enhancers, crucial for cell identity and the aberrant expression of genes that promote cancer cell fitness, may be involved in this process [143]. Given that the super-enhancer landscape undergoes significant reprogramming in cancer, exploring differential enhancers active in tumors can shed light on how epigenetic events may contribute to the dysregulation of FST in HNSCC. This knowledge is particularly significant given that chromatin-remodeling events are reversible and have become the focus of numerous ongoing clinical trials [144].

Additional factors influencing *FST* expression have also been documented in the literature. In the context of HNSCC, Chen et al. show that the frequently altered member of the cadherin-like protein family, FAT1, promotes the nuclear translocation and functional activation of YAP1. Interestingly, YAP1, a HIPPO pathway effector, is frequently upregulated in HNSCC and has been shown to bind the promoter region of the *FST* gene (Figure 6C), leading to tumorigenesis and the suppression of immune responses [145]. Importantly, the inhibition of BRD4, a chromatin remodeler, impairs YAP1 recruitment, consequently leading to the downregulation of FST. This suggests that BET protein inhibitors, such as JQ1, could present a potential strategy for manipulating FST levels in tumors.

Intriguingly, YAP1 also functions as a coregulator in an immune evasion program orchestrated by TEAD/p63 within airway basal epithelial cells, where *FST* serves as a target gene [146]. These findings, along with the positive association between increased *FST* expression in HNSCC and tumor-infiltrating immunosuppressive cells as reported by our group [126], further support the role of transcriptional cross-talk that acts upstream of *FST* in fostering an immune-evasive TME (Figure 3B and Figure 6). Together, these discoveries provide novel insights into FST regulation, particularly by oncogenes, and offer a potential mechanism for its dysregulation in squamous cancers, which intersects with its positive regulation by TGF-β [139,147].

Collectively, the evidence surrounding FST in lung, ovarian, and HNSCC underscores the role of FST as an intermediary between tumor epithelial cells and the stroma, where it facilitates the aggressiveness of cancer cells, likely by fostering an immune-cold and drug-resistant TME.

## 10. Concluding Remarks and Future Prospects

FST is critical for normal development and tissue homeostasis, as evident by the perinatal death and broad multi-organ phenotypes of FST knockout mice [148]. This function of FST is further substantiated by exome sequencing-based studies revealing a mutant variant of FST that impedes interaction with GDF11, a TGF-β family member, resulting in orofacial clefts in human patients akin to FST-null mice [149]. Moreover, gain-of-function studies illustrate FST’s capacity to override TGF-β signals, inducing cellular growth, delaying differentiation, and promoting metabolic reprogramming [150,151,152]. Notably, in humans, elevated FST is linked to metabolic disorders such as type II diabetes and cancer [42,67]. While these results and observations underscore FST’s growing importance in regulating both normal developmental programs and disease states, the role of the FST–TGF-β axis in the progression and therapeutic resistance of cancer has particularly gained prominence and experimental scrutiny in recent years.

Patients of various cancer types demonstrate increased levels of FST in their serum, making FST a potential biomarker [63,68,153,154]. Furthermore, the source of this increased FST is believed to originate from solid tumors, with transcriptomic studies revealing elevated expression of *FST* in tumor tissues [38,86,126]. Functional studies in preclinical model systems and patient data analyses support a regulatory role of FST in various oncogenic processes; cell proliferation, stem cell renewal, metastasis, and angiogenesis are prominent among them. Notably, dysregulated FST in cancer cells has emerged as a critical factor in promoting tumor growth, an immunosuppressive TME, and, importantly, resistance to commonly utilized cancer drugs. While TGF-β pathway inhibitors have been the long-term focus of anticancer agents, currently existing therapies have not been as successful [133]. This underscores the need for a closer examination of FST as a potent and effective TGF-β antagonist that can be used to strategically target or prevent cancer cells from becoming resistant to the anti-proliferative effects of TGF-β and also to serve as a biomarker that can better stratify cancer patients and improve their responses to therapy.

Given the current focus on secreted molecules in drug discovery, understanding the function of FST in individual diseases holds promise for developing novel therapies. Indeed, FST has emerged as a drug candidate for muscle wasting diseases, with preclinical and clinical trials showcasing its therapeutic potential via gene delivery, fusion proteins, and nanoparticle-based mRNA therapy [29,51,155,156,157,158,159]. Further investigations are needed to elucidate the functional role of FST in tumor biology and whether it presents a candidate target for small-molecule inhibition or is itself a molecule to enhance therapeutic response, ultimately improving health outcomes for cancer patients.

It is becoming clear that epithelial cells are one major source of *FST* dysregulation in tumors [126]. However, whether this arises from a TGF-β-stimulated immune onslaught or whether CAFs play a role warrants further investigation. Critical insights into mechanisms influencing FST expression and its function are likely to shed light on how the TME is reprogrammed during oncogenesis and therapeutic resistance. Additionally, future cancer-centric experimental designs need to consider the diverse biochemical properties of FST, such as a differing affinity for TGF-β ligands, the relative abundance of varying FST isoforms, and post-translational variants. Such studies are likely to have a major and far-reaching impact on not only the cancer types discussed in this review, but also on other tumors in which FST has been relatively understudied.

## Figures and Tables

**Figure 1 biology-13-00130-f001:**
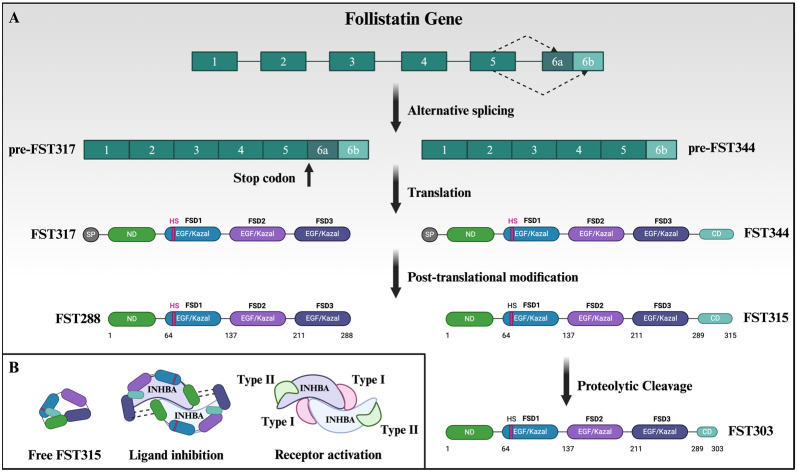
Organization and function of the FST domains. The follistatin gene (*FST*) consists of six exons, each encoding specific domains (**A**). Exon 1 encodes the signal peptide, while exon 2 codes for the N-terminal domain. Exons 3–5 encode the domains FSD1 to -3, and alternative splicing events at exon 6 generate two pre-FST mRNA molecules. Exon 6 contains a stop codon, resulting in the translation of the shorter FST317 protein lacking the C-terminal domain, whereas exon 6b yields the longer FST344 isoform. Post-translational cleavage removes the 29-amino-acid signal peptide from both FST317 and FST344, resulting in isoforms FST288 and FST315, respectively. The follicular isoform, FST303, is derived from the proteolytic cleavage of a portion of the C-terminal domain of FST315. The positive charge of the heparin sulfate (HS) binding domain within FSD1 interacts with the acidic C-terminal domain in apo-FST315 (**B**). This interaction prevents the association of apo-FST315 with the cell surface. In circulation, FST315 binds to transforming growth factor-β (TGF-β) members such as activin, which is composed of inhibin β-subunits, in a 2:1 stoichiometry. This binding occludes the types I and II receptor binding sites of activin. In the absence of FST, activin forms a complex with two type I and type II serine/threonine kinase receptors. The N-terminal, FSD1, and FSD2 domains of FST are essential for this interaction. In the case of FST315, this interaction frees the HS binding site, enabling the association between the FST–activin complex and the cell surface. This facilitates the subsequent internalization and degradation of TGF-β ligands. Created with BioRender.com (accessed on 17 February 2024).

**Figure 2 biology-13-00130-f002:**
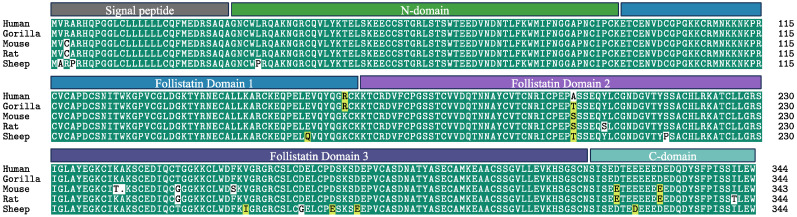
FST is highly conserved among species. FST sequence alignment demonstrates a sequence similarity of over 90% between human, gorilla, mouse, rat, and sheep species. Human and mouse FST is 98% identical, thereby rendering mouse models highly applicable for the study of FST biology and any potential drug candidates targeting FST. Non-highlighted amino acids are non-conserved, while amino acids highlighted in yellow represent similar residues; green represents conserved sequences. Figure created with R package ‘msa’ (accessed on 2 January 2024).

**Figure 3 biology-13-00130-f003:**
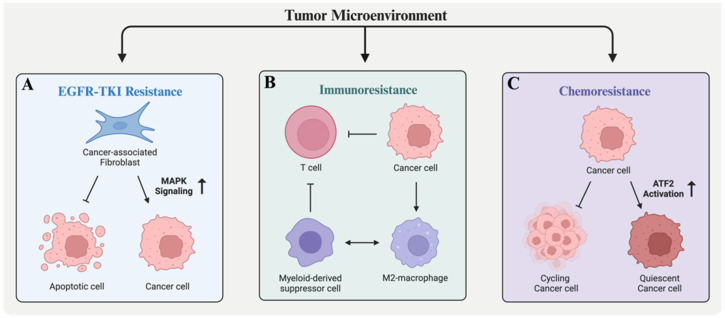
Mechanisms of FST-induced drug resistance. Cross-talk between cancer cells and the surrounding microenvironment contributes to therapeutic resistance. In lung cancer, FST is one of the top factors secreted by cancer-associated fibroblasts (CAFs) that protect cancer cells from epidermal growth factor–tyrosine kinase inhibitor (EGFR-TKI)-induced cytotoxicity (by independently activating the MAPK pathway) (**A**). The overexpression of FST in ovarian and head and neck cancers is associated with reduced inflammatory responses and increased myeloid-derived suppressor cell infiltration and may thus explain the reduced efficacy of immunotherapies observed in mouse models (**B**). Additionally, platinum-based chemotherapy induces FST expression in ovarian cancer, leading to paracrine induction of cellular quiescence and reduced cell death via increased phosphorylation of the stress pathway effector activating transcription factor 2 (ATF2) (**C**). Created with BioRender.com (accessed on 12 December 2023).

**Figure 4 biology-13-00130-f004:**
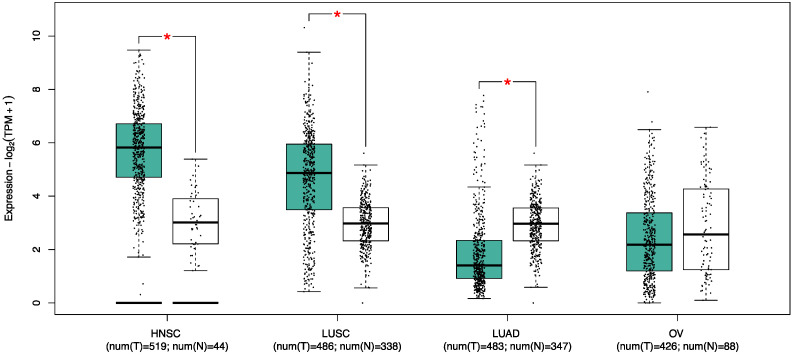
Dysregulation of FST in squamous cell carcinoma. *FST* expression is upregulated in head and neck squamous cell carcinoma (HNSC) and lung squamous cell carcinoma (LUSC) compared to normal tissues, while *FST* is downregulated in lung adenocarcinoma (LUAD). No statistical difference is observed in ovarian cancer (OV); however, these data originate from ovarian serous cystadenocarcinoma from the Genotype-Tissue Expression Portal, as The Cancer Genome Atlas lacks comparable normal data for OV. Expression of *FST* is shown in transcripts per million (TPM); normal tissue, white; tumor tissue, green. * *p*-value < 0.05, one-way ANOVA. Data were obtained with GEPIA (accessed on 8 November 2023).

**Figure 5 biology-13-00130-f005:**
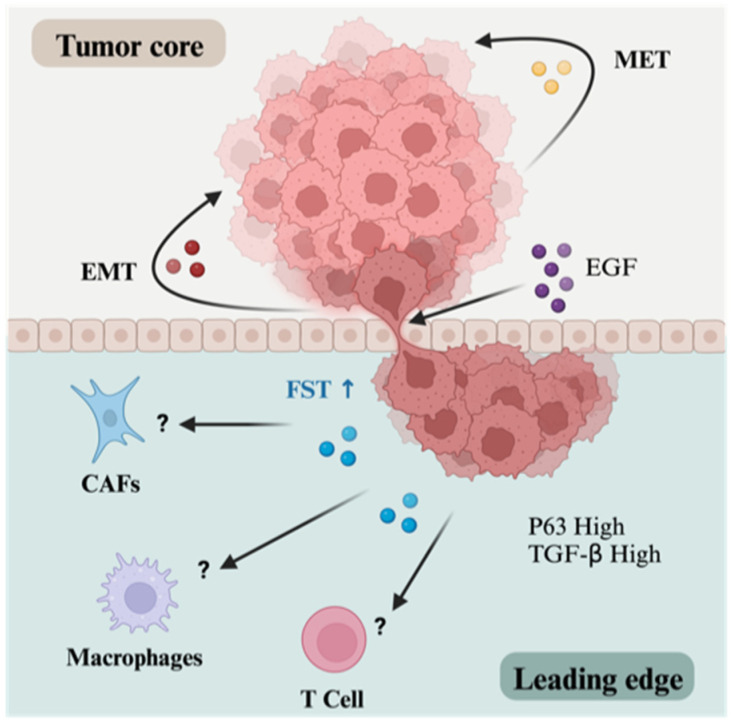
Enrichment of FST within the tumor leading edge. HNSCC tumors can be spatially dichotomized based on clinicopathological and signaling features into the tumor core and leading edge. Whereas cells of the tumor core are maintained in a non-invasive state via mesenchymal-to-epithelial transition (MET) signaling, leading edge cells engage with the tumor core to induce malignant phenotypes via epithelial-to-mesenchymal transition (EMT) signaling. Within this niche, oncogene *TP63* is upregulated, and TGF-β signaling dominates outgoing signals. Remarkably, FST is also enriched within the leading edge. The upregulation of FST within these cells may be mediated by growth factors derived from the tumor microenvironment (TME), such as EGF. Downstream activation of the MAPK pathway by EGF drives a p63 transcriptional program potentially responsible for FST dysregulation in HNSCC. Unfortunately, the impact of this dysregulation on important cells of the TME known to impact tumor behavior and therapy response, CAFs, and immune cells such as macrophages and T cells remains to be determined (represented by ?). Created with BioRender.com (accessed on 13 January 2024).

**Figure 6 biology-13-00130-f006:**
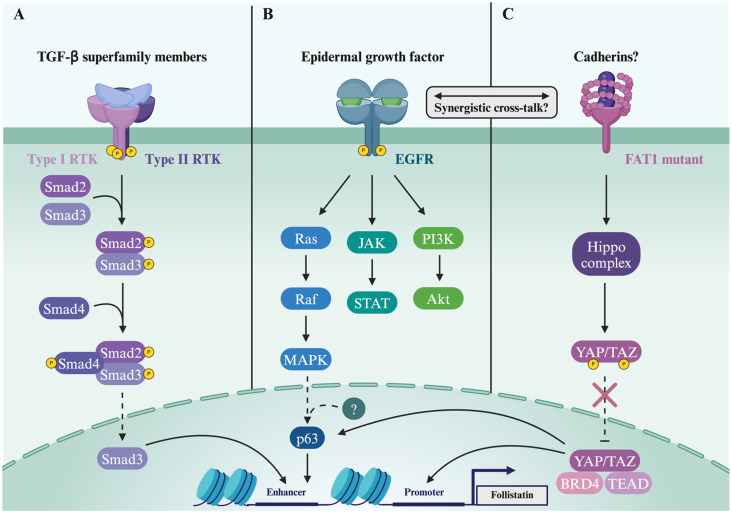
Signaling pathways driving *FST* in cancer. TGF-β regulates its own bioavailability via an auto-regulatory feedback loop that drives *FST* expression through the activation and nuclear translocation of Smad mediators that bind an *FST* enhancer (**A**). Epidermal growth factor (EGF) signaling via the MAPK pathway also regulates *FST* expression via the recruitment of p63 and unknown cofactors to an upstream enhancer (**B**). FAT1 is a cadherin involved in multiple protein–protein interactions and is one of the most mutated genes in HNSCC. Inactivating mutations of FAT1 causes the inactivation of the Hippo pathway, which would otherwise keep YAP1 phosphorylated and inactive. This leads to YAP1 nuclear translocation where it recruits coactivators TEAD and the chromatin remodeler BRD4 to access the promoter region of *FST* (**C**). FAT1 potentially interacts with cadherins to regulate cellular polarity and actin dynamics but endogenous ligands for FAT1 are still under investigation. Although a strong correlation between mutant FAT1 and EGF pathway activation has been established, it is yet unknown whether mutant FAT1 directly interacts with the EGF receptor or whether YAP1 regulation leads to increased EGF activity. Lastly, the convergence of YAP/TAZ, TEAD, and TP63 activity has also been shown to drive the progression of bronchial premalignant lesions via the regulation of genes such as *FST.* Created with BioRender.com (accessed on 5 January 2024).

## Data Availability

Not applicable.

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
