# Peer review of "The Reign of Follistatin in Tumors and Their Microenvironment: Implications for Drug Resistance"

_biology, 2024, doi:10.3390/biology13020130_

Round 1
Reviewer 1 Report
Comments and Suggestions for Authors
The study investigates the role of Follistatin (FST) in lung, ovarian, and Head and Neck Squamous Cell Carcinoma (HNSCC), emphasizing its impact on tumor microenvironments. FST exhibits complex, context-dependent effects on immune responses, inflammatory signals, and therapeutic resistance in cancer. The manuscript provides insights into FST dysregulation across diverse cancers, suggesting its potential as a therapeutic target and highlighting the need for further exploration of its clinical implications. Some comments are listed below for authors considerations:
Clarity:
- The manuscript transitions abruptly between different topics and diseases, such as lung cancer, ovarian cancer, and head and neck squamous cell carcinoma, without clear delineation.
- The introduction of various concepts, such as the role of FST in different cancer types and its impact on treatment outcomes, is scattered throughout the text, making it difficult for readers to grasp a coherent narrative.
- The section discussing mechanisms driving FST expression in cancer lacks a smooth transition and appears disconnected from the preceding sections, impacting the overall flow of the manuscript.
- The figure captions (e.g., Figure 5) provide information without sufficient context, contributing to confusion regarding the relevance of the figures to the text.
Redundancy:
- The information about FST's role in different cancer types is repeated in multiple sections without offering new insights, creating redundancy in the manuscript.
- The statement "This suggests that FST may exhibit opposing roles in adaptive and innate immune responses" is reiterated later without providing additional context or elaboration.
- The discussion on the impact of FST on treatment outcomes, particularly in lung cancer, is presented in multiple sections with similar wording, contributing to unnecessary repetition.
- The mention of FST's potential protective function in the tumor microenvironment is repeated without introducing fresh perspectives or relevant details.
- Certain phrases, such as "dysregulation of FST," are used repetitively without offering nuanced explanations or connecting them seamlessly to the broader context.
- The information about FST upregulation in specific cancers, like head and neck squamous cell carcinoma, is repeated in different sections without introducing novel aspects or clarifications.
Data presentation:
Fully understanding the provided text contains rich information on gene expression, pathways, and experimental results. However, the presentation of this data may be overwhelming for readers due to its extensive nature. To improve the accessibility of complex information, it is advisable to consider incorporating figures, tables, and visual aids strategically throughout the manuscript. For example, what is the benefit of Figure 2 very difficult to follow with no clear legend it should move to supplementary data
Figures:
To improve this aspect, it is essential to ensure that figures are appropriately cited within the text. Each figure should be accompanied by clear and informative captions that explicitly explain their relevance to the ongoing discussion. By doing so, readers can easily understand the connection between the textual content and the visual elements, enhancing the overall clarity and impact of the manuscript.
Example:
- "As shown in Figure 1, the gene expression levels exhibit a significant increase under treatment conditions. This upregulation correlates with the observed phenotypic changes discussed in the preceding paragraphs."
Figure legend is very small and barely readable. some of the inner writing which generated by biorender is distorted to be able to read clearly
Statements lack citations:
- Biomarker analysis revealed elevated levels of FST in the sera of patients subjected to combination therapy."
- "NSCLC is a heterogeneous disease including the more prevalent subtypes, lung adenocarcinoma (LUAD) and lung squamous cell carcinoma (LUSCC)."
- "Inhibition of TGF-β signaling via systemic administration of FST increases the efficacy of carboplatin therapy in LUAD."
- "FST may play a more significant role in lung squamous cell carcinoma (LUSCC) compared to lung adenocarcinoma (LUAD)."
- "The release of FST by cancer-associated fibroblasts (CAFs) forms a protective barrier around tumor cells, making them less responsive to TGF-β signals."
- "FST is upregulated in head and neck squamous cell carcinoma (HNSC) and lung squamous cell carcinoma (LUSC) compared to normal tissue."
- "Elevated FST is associated with a shorter overall survival in HNSC and LUAD.
-
Explanation of Molecular Mechanisms:
- The manuscript delves into molecular mechanisms involving pathways such as MAPK signaling, but the explanations are often brief.
- Expand on the molecular mechanisms, providing detailed explanations and potentially incorporating schematic diagrams for clarity.
-
Clinical Implications:
- The manuscript mentions the clinical relevance of FST in cancer but lacks a comprehensive exploration of potential clinical implications and applications.
- Develop a more thorough discussion on how the findings may translate into clinical applications or inform future research.
-
Discussion of Limitations:
- The manuscript could benefit from a more explicit discussion of its limitations and potential sources of bias.
- Address any methodological limitations, sample size considerations, or other factors that might impact the robustness of the findings.
-
Integration of Findings Across Cancers:
- While the manuscript discusses FST in the context of lung, ovarian, and HNSCC, there could be more integration and comparison of findings across these different cancers.
- Emphasize commonalities and differences, providing a more holistic view of FST's role in various cancer types.
-
Recommendations for Future Research:
- The manuscript lacks a section that clearly outlines recommendations for future research.
- Include a dedicated section suggesting potential directions for future studies based on the current findings.
Comments on the Quality of English Language
The language used in the writing is generally clear and scientifically appropriate. However, there are instances where the text could benefit from improved clarity and conciseness. Some sentences are lengthy and complex, potentially hindering readability. Additionally, careful attention to grammar and punctuation throughout the manuscript could enhance overall fluency.
Author Response
Reviewer 1
The study investigates the role of Follistatin (FST) in lung, ovarian, and Head and Neck Squamous Cell Carcinoma (HNSCC), emphasizing its impact on tumor microenvironments. FST exhibits complex, context-dependent effects on immune responses, inflammatory signals, and therapeutic resistance in cancer. The manuscript provides insights into FST dysregulation across diverse cancers, suggesting its potential as a therapeutic target and highlighting the need for further exploration of its clinical implications. Some comments are listed below for authors considerations:
Clarity:
The manuscript transitions abruptly between different topics and diseases, such as lung cancer, ovarian cancer, and head and neck squamous cell carcinoma, without clear delineation.
The introduction of various concepts, such as the role of FST in different cancer types and its impact on treatment outcomes, is scattered throughout the text, making it difficult for readers to grasp a coherent narrative.
The section discussing mechanisms driving FST expression in cancer lacks a smooth transition and appears disconnected from the preceding sections, impacting the overall flow of the manuscript.
We understand the concerns regarding the transition between sections. We have addressed this by adding several transition segments and moving some of the segments around and readjustment of the figures (such as discussing mechanisms driving FST expression in cancer). We hope that these alterations now allow a better flow of the topics and addresses the disconnection that was perceived in the previous iteration. We thank the reviewer for this suggestion since it has in our humble view significantly improved the manuscript.
The figure captions (e.g., Figure 5) provide information without sufficient context, contributing to confusion regarding the relevance of the figures to the text.
The figure in question provided expression data for FST across the different cancers discussed in the text along with survival analysis curves. In the revised version (now figure 4), the Kaplan-Meier plots were removed as to only show the expression data. The context in the figure legend is as discussed in the main text – to emphasize the different levels of FST expression between normal and tumor tissue that fits well with a potential oncogenic function for FST.
Redundancy:
The information about FST's role in different cancer types is repeated in multiple sections without offering new insights, creating redundancy in the manuscript.
This has been addressed by limiting the mention of FST in the cancer context outside of the appropriate sections (i.e., cancer hallmarks and drug resistance).
The statement "This suggests that FST may exhibit opposing roles in adaptive and innate immune responses" is reiterated later without providing additional context or elaboration.
This has been addressed by removing the redundant sentences.
The discussion on the impact of FST on treatment outcomes, particularly in lung cancer, is presented in multiple sections with similar wording, contributing to unnecessary repetition.
This has been addressed by removing unnecessary repetition.
The mention of FST's potential protective function in the tumor microenvironment is repeated without introducing fresh perspectives or relevant details.
This has been addressed by limiting redundant sentences that don’t offer additional perspectives.
Certain phrases, such as "dysregulation of FST," are used repetitively without offering nuanced explanations or connecting them seamlessly to the broader context.
This has been considered and the manuscript updated to limit excessive repetition of the word “dysregulation”. We have opted to use “increased levels of FST” for example, as the appropriate term in the proper context when applicable.
The information about FST upregulation in specific cancers, like head and neck squamous cell carcinoma, is repeated in different sections without introducing novel aspects or clarifications.
This concern has been addressed by limiting discussions of FST in the head and neck cancer outside of the cancer section. We must stress that many of the recent findings about FST has been in the context of head and neck squamous carcinoma – hence this topic might seem more redundant.
Data presentation:
Fully understanding the provided text contains rich information on gene expression, pathways, and experimental results. However, the presentation of this data may be overwhelming for readers due to its extensive nature. To improve the accessibility of complex information, it is advisable to consider incorporating figures, tables, and visual aids strategically throughout the manuscript. For example, what is the benefit of Figure 2 very difficult to follow with no clear legend it should move to supplementary data
The benefit of figure 2 is to show the remarkable conservation of FST amino acid sequences between species which is relevant for translational research. This is further discussed in our revision. In figure 6, we show summary of key pathways discussed throughout the text.
Figures:
To improve this aspect, it is essential to ensure that figures are appropriately cited within the text. Each figure should be accompanied by clear and informative captions that explicitly explain their relevance to the ongoing discussion. By doing so, readers can easily understand the connection between the textual content and the visual elements, enhancing the overall clarity and impact of the manuscript.
Example:
"As shown in Figure 1, the gene expression levels exhibit a significant increase under treatment conditions. This upregulation correlates with the observed phenotypic changes discussed in the preceding paragraphs."
We have taken this into account and reworked the order of our figures within the text to enhance the flow.
Figure legend is very small and barely readable. some of the inner writing which generated by biorender is distorted to be able to read clearly
The size of the figure legend font is set by the journal. We have made changes to some figures to ensure that the writings are legible and clear
Statements lack citations:
Biomarker analysis revealed elevated levels of FST in the sera of patients subjected to combination therapy."
"NSCLC is a heterogeneous disease including the more prevalent subtypes, lung adenocarcinoma (LUAD) and lung squamous cell carcinoma (LUSCC)."
"Inhibition of TGF-β signaling via systemic administration of FST increases the efficacy of carboplatin therapy in LUAD."
"FST may play a more significant role in lung squamous cell carcinoma (LUSCC) compared to lung adenocarcinoma (LUAD)."
"The release of FST by cancer-associated fibroblasts (CAFs) forms a protective barrier around tumor cells, making them less responsive to TGF-β signals."
"FST is upregulated in head and neck squamous cell carcinoma (HNSC) and lung squamous cell carcinoma (LUSC) compared to normal tissue."
"Elevated FST is associated with a shorter overall survival in HNSC and LUAD.
We have incorporated references as needed for some of the sentences that have been by the reviewer. Note that some of the sentences being asked to be referenced are general statements and not results being discussed based on a publication. The last two sentences are in reference to our figure and based on publicly available TCGA data.
Explanation of Molecular Mechanisms:
The manuscript delves into molecular mechanisms involving pathways such as MAPK signaling, but the explanations are often brief.
Expand on the molecular mechanisms, providing detailed explanations and potentially incorporating schematic diagrams for clarity.
The molecular mechanisms that are relevant to our discussion of FST biology have been summarized in figure 6. The MAPK pathway was mentioned in several instances as a pathway that may be regulated by FST and it itself regulating FST expression. We have not delved deeper into molecular mechanisms, because that is precisely what is lacking in the field of FST biology. Any discussion on such topic would be merely speculative, hence we have refrained from doing so.
Clinical Implications:
The manuscript mentions the clinical relevance of FST in cancer but lacks a comprehensive exploration of potential clinical implications and applications.
Develop a more thorough discussion on how the findings may translate into clinical applications or inform future research.
The evidence presented in this manuscript suggest that there may be a potential for FST in clinical settings as a biomarker to stratify patients. We summarize the data and future prospects in our revised discussion section (emphasized in the Conclusion segment).
Discussion of Limitations:
The manuscript could benefit from a more explicit discussion of its limitations and potential sources of bias.
Address any methodological limitations, sample size considerations, or other factors that might impact the robustness of the findings.
There are no limitations or potential sources of bias to report.
Integration of Findings Across Cancers:
While the manuscript discusses FST in the context of lung, ovarian, and HNSCC, there could be more integration and comparison of findings across these different cancers.
Emphasize commonalities and differences, providing a more holistic view of FST's role in various cancer types.
We have made comparison between diseases within the manuscript but chose not to elaborate on them to ensure brevity and the right flow of the topics covered in this Review.
Recommendations for Future Research:
The manuscript lacks a section that clearly outlines recommendations for future research.
Include a dedicated section suggesting potential directions for future studies based on the current findings.
We thank the reviewer for this useful suggestion. We have now revised our discussion section to suggest specific examples of future potential studies that would help address many unanswered questions surrounding FST.
Comments on the Quality of English Language
The language used in the writing is generally clear and scientifically appropriate. However, there are instances where the text could benefit from improved clarity and conciseness. Some sentences are lengthy and complex, potentially hindering readability. Additionally, careful attention to grammar and punctuation throughout the manuscript could enhance overall fluency.
This has been taken into consideration and the manuscript was updated accordingly by removing run-on sentences and incorporating appropriate punctuation. We thank the reviewer for their helpful suggestions.

Reviewer 2 Report
Comments and Suggestions for Authors
This is a comprehensive, informative and very well written review article and I only have some minor suggestions for consideration.
1. Given the high degree of homology of FST across species, it would be helpful to indicate whether TGF beta neutralisation by FST is also conserved across species (e.g. do human orthologs neutralise mouse TGF-beta ligands and vice versa).
2. Line 270 - Perhaps I have misunderstood but should this sentence read "One interesting possibility is that FST contributes to the dichotomous nature of TGF-β signaling in cancer by promoting resistance to the cytostatic effects of TGF-β in tumor cells."
3. As a general point the authors note the biphasic effects of TGF beta family members in cancer, with a preponderance of tumor inhibitory effects in early disease and tumor promoting effects later on. Can the authors comment on whether similar biphasic effects apply to FST, given the evident complexity of the system.
4. Is there any evidence that the FST system exerts TGF beta independent effects?
Author Response
Reviewer 2
This is a comprehensive, informative and very well written review article and I only have some minor suggestions for consideration.
- Given the high degree of homology of FST across species, it would be helpful to indicate whether TGF beta neutralisation by FST is also conserved across species (e.g. do human orthologs neutralise mouse TGF-beta ligands and vice versa).
This is an excellent point made by reviewer. Recombinant human FST has been administered to mice across many studies, for e.g., showing increased muscle mass, a well-known effect of myostatin inhibition by FST. Moreover, several studies also report increased muscle mass in mice overexpressing human FST. Additionally, TGF-Bs are also highly conserved proteins. We have added this specific point within the manuscript and discuss the cross-species conserved biology and activity of FST.
- Line 270 - Perhaps I have misunderstood but should this sentence read "One interesting possibility is that FST contributes to the dichotomous nature of TGF-β signaling in cancer by promoting resistance to the cytostatic effects of TGF-β in tumor cells."
Thank you for pointing out this error. This indeed should have read “cytostatic” and not “anti-cytostatic.” We have corrected this in the manuscript.
- As a general point the authors note the biphasic effects of TGF beta family members in cancer, with a preponderance of tumor inhibitory effects in early disease and tumor promoting effects later on. Can the authors comment on whether similar biphasic effects apply to FST, given the evident complexity of the system.
As the reviewer correctly surmises, it is possible that FST also has a biphasic role in cancer mirroring that of TGFB, however this has not been explored. We suspect that elevated FST in late-stage cancers must have some consequences on TGFB signaling possibly creating a negative feedback loop. We contend that animal models and single cell analysis could be quite useful in deciphering the biphasic nature of FST, if any during cancer progression. More work is needed to really tease this out – hence we have decided to not broadly comment on this aspect of FST biology, given the lack of experimental data.
- Is there any evidence that the FST system exerts TGF beta independent effects?
There is some evidence in adipose tissue where FST has been shown to upregulate the PIKT pathway and promote free fatty acid production and in ovarian cancer where FST has been shown to activate p-ATF2 to promote chemoresistance. This does not necessarily mean that these effects are completely independent of TGF-B; as canonical and non-canonical pathways activated by TGF-B can often intersect. We suspect pleiotropic effects of FST that interface with other signaling pathways are possible, but decided to focus on the TGF-Beta angle for this Review given this is the primary pathway that in inhibited by FST as evident by a well-documented literature.

Round 2
Reviewer 1 Report
Comments and Suggestions for Authors
No further comments!